# Development and Properties of Fish Gelatin/Oxidized Starch Double Network Film Catalyzed by Thermal Treatment and Schiff’ Base Reaction

**DOI:** 10.3390/polym11122065

**Published:** 2019-12-11

**Authors:** Yuhao Dong, Hao Chen, Peng Qiao, Zijing Liu

**Affiliations:** 1Marine College, Shandong University (Weihai), Weihai 264209, China; dongyh97@163.com (Y.D.);; 2College of Food Science and Technology, South China University of Technology, 381 Wushan Road, Guangzhou 510640, China; 3Beijing Advanced Innovation Center for Food Nutrition and Human Health, Beijing Technology and Business University (BTBU), Beijing 100037, China

**Keywords:** fish gelatin, oxidized starch, Schiff’ base reaction, composite film, double network

## Abstract

In order to improve the properties of fish gelatin (FG), oxidized starch (OS) was adopted to form hetero-covalent linkage with it based on thermal treatment and the Schiff’ base reaction. The effects of different ratios of FG/OS (ranging from 10:1 to 2:1) on the properties of films were investigated. OS improved the mechanical and barrier properties of films significantly, while the moisture content decreased as OS concentration increased. The optimum concentration was obtained at the loading amount of 1.5% (w/v) OS. FT-IR spectra revealed the covalent cross-linking between FG and OS induced by Schiff’ base reaction. Moreover, composite films had superior preservation effect on blueberry, according to the results of weight loss, total soluble solids, titratable acidity, and total anthocyanin content. Therefore, this study suggested that FG-OS double network films (FODF) has great potential in the packaging industry.

## 1. Introduction

In the past few decades, a great deal of work has been conducted on edible films. The reasons could be mainly attributed to the pursuit for high quality and safety food, the need of innovative preservation techniques to meet new product requirements as well as the concerns about environmental pollutions [1,2]. They are usually developed with renewable by-products from agricultural or marine sources, replacing traditional plastic polymers obtained from fossil resources [3].

Proteins, polysaccharides, lipids, or their composites are most adopted to fabricate edible films. In particular, protein film has remarkable gas barrier properties, while its physical and mechanical properties are inferior to polysaccharides. Lipid film shows low water vapor permeability. Since each material owns its unique properties, including advantages and drawbacks, to blend them and comprehensively utilize the materials becomes an effective approach to enhance the final performance [4].

As a kind of novel film forming material, protein-polysaccharide composites films have been widely studied. The presence of carboxymethyl konjac glucomannan could enhance the tensile strength and elongation at break of soy proteins isolate according to Le Wang et al. [5]. Zunying Liu et al. found that the glass transimation temperature of chitosan-gelatin film progressively increased with increasing chitosan level [6]. Gelatin-glucose films based on Maillard reaction showed favorable antioxidant capacity [3]. However, most of these membrane structures currently under study are transient network that are maintained by physical effects, including electrostatic action, Van der Waals force, hydrogen bonding, etc. The combing process would be influenced by many parameters such as salts, temperature, and pH values. Compared to that, covalent bond has more permanent and firmer structure [7].

Protein-polysaccharide conjugates could be obtained in different ways, such as Maillard reaction, enzymatic catalysis, etc. [8]. As a kind of non-enzymatic browning reactions, Maillard reaction can produce brown substances, [3], which would affect the aesthetics of the package. Enzymes such as laccase and peroxidase have been supplied to covalently cross-link proteins and hydrocolloids for a long time, but the cost should be enhanced to some extent [8].

In the current research, fish gelatin (FG) and oxidized starch (OS) were supplied as the film forming substrates. Starch is a popular food material with good film forming properties, especially in consideration of cost, environment, and safety. [9]. However, previous studies have shown that the mixing of gelatin and starch results in the phase separation, which would affect the rheological, processing, and mechanical properties of the mixture [10]. Research showed that the compatibility of starch and gelatin can be improved by the oxidative modification of the starch to increase the content of the carboxyl and carbonyl groups in the starch. The cross-linking between OS and gelatin can improve the microstructure. [11].

OS-FG interpenetrating and cross-linking network were successively fabricated by thermal treatment and Schiff’ base reaction. The aldehyde groups in oxidized starch could crosslink with amino groups of gelatin by imine group (-C = N-) linkages and form hetero-covalent linkages polymers [12]. Appendix A illustrated the mechanism of Schiff’ base reaction between gelation and oxidized starch. Thus, an improved understanding of the kinetics of the individually steps will lead to better possibilities for film formation with superior mechanical, barrier, and fresh-keeping performance.

## 2. Materials and Methods

### 2.1. Materials and Reagents

Fish gelatin (250 bloom value) was supplied by Vinhwellness Co., Ltd. (Cao Lanh, Vietnam). Oxidized starch (05548) was a kind gift from Cargill Asia Pacific Food Systems (Beijing) Co., Ltd. Glycerol were purchased from Hongyan Reagent Factory (Tianjin, China). Other chemicals and reagents used were of analytical grade.

### 2.2. Preparation of Fish Gelatin/Oxidized Starch Composite Film

The fabrication process was performed as follows: Firstly, FG and OS were separately dissolved in deionized water by magnetic stirring for 5 h at room temperature. Subsequently, the fully dissolved FG and OS solutions were heated at 80 °C for 30 min respectively. Previous work (Appendix A) indicated that the best properties of FG film were obtained at the concentration of 5% (w/v). The blend solutions were prepared at different concentration of 0, 0.5, 1.0, 1.5, 2.0, 2.5% (w/v) OS and 5% (w/v) FG. They were referred to as FG, FO1, FO2, FO3, FO4, and FO5, respectively. Then, the solutions were heated at 50 °C for 4 h. 3% (w/v) glycerol was added as plasticiser under stirring continuously. Finally, the film-forming dispersion was cast onto a polyethylene plate (90 mm × 90 mm) and dried at 30 ± 2 °C for 24 h [13]. All films were stored in the shade of the room at room temperature and 60% RH for 24 h prior performing the further analysis.

### 2.3. Fourier Transform Infrared (FT-IR) Spectroscopy

FT-IR spectra of the films were recorded on an infrared spectrometer (Thermo Nicolet iS50, Thermo Fisher Scientific, Waltham, MA, USA) in the defined range of 4000–400 cm^−1^. All spectra were recorded using a 0.4 cm^−1^ resolution with 30 scans. [12].

### 2.4. Physico-Chemical Properties of Films

#### 2.4.1. Color Values

The color of films was measured using a colorimeter (NR110, Shenzhen Sanenchi technology Co., Ltd., Shenzhen, China). The CIE Lab scale chromaticity parameters (*L**: lightness/brightness, *a**: redness/greenness, and *b**: yellowness/blueness) were recorded. The color values of FO1 was set as the control to calculate total color difference (*ΔE*). *ΔE* was calculated according to the following equation [14]:(1)$$\Delta E=\sqrt{{\left(L*-L\right)}^{2}+{\left(a*-a\right)}^{2}+{\left(b*-b\right)}^{2}}$$
where *L*, *a*, and *b* were the color parameter values of film samples, *L**, *a**, and *b** were the color parameter values of the control film (FO1).

#### 2.4.2. Film Thickness

A digital micrometer (211-101, Dongguan Jingyou Mould hardware Co. Ltd., Dongguan, China) equipped with accuracy scale of 0.001 mm was used to determine thickness. Measurements were randomly performed on five different positions of each film.

#### 2.4.3. Moisture Content (MC)

The samples were exposed in a hot air oven dryer (105 °C) for 24 h until reaching constant weight. The MC was calculated according to the below equation:(2)$$MC=\frac{\left(m_{2}-m_{3}\right)}{m_{1}}\times 100\%$$
where *m*_1_ was the initial weight of film, *m*_2_ was the weight of film and discs before being dried, and *m*_3_ was the constant weight of film and discs after being dried at 105 °C.

#### 2.4.4. Water Vapor Permeability (WVP)

WVP was determined according to the ASTM, E96-80 Standard test method with some modifications [15]. Filling 9 g anhydrous CaCl_2_ in the permeation cups (50 mm height and 10 mm inner diameter), then, covering the films (diameter: 30 mm) onto the cups and sealing with vaseline. Eventually, they were placed in desiccators containing saturated solution of KBr (60% RH) at room temperature. The changes in the cell weight were recorded for 4 days. WVP was calculated as follows:(3)$$WVP=\frac{\Delta m\times d}{A\times t\times \Delta P}$$
where Δm was the weight gain (g), *d* was the film thickness (m), *A* was the valid permeation area (m^2^), *t* was the permeation time (d), and ΔP was the partial vapor pressure difference between both sides of the membrane.

#### 2.4.5. Permeability to Oil

Firstly, the films were accurately sealed on the test tube loaded with 5 mL soybean oil. Then, the tube was inverted and placed on a piece of filter paper. The complete set of equipment was stored at room temperature in desiccators containing saturated solution of KBr (60% RH) for 24 h [16]. Oil permeability was calculated by the following equation:(4)$$Oil\;Permeability=\frac{\Delta m}{A\times t}$$
where Δm was the weight variation of the filter paper (g), *A* was the area of effective contact (m^2^), and *t* was the storage period (h).

#### 2.4.6. Film Permeability to Oxygen

The oxygen permeability (OP) was analyzed using an Ox-Tran system (VAC-V1, Mocon, Minneapolis, MN, USA) at 53% RH and 25 °C. The film samples were placed on the sample chambers with a round area 4.9 cm^2^ exposed for measurement, oxygen flowed in one side of the films and nitrogen flowed in the other side [17].

#### 2.4.7. Light Transmission and Transparency of Films

The light barrier properties of films against visible light and ultraviolet (UV) were determined using a UV-Visible spectrophotometer (T6, Beijing Puxi General instrument Co., Ltd., Beijing, China) at the selected wavelengths (200–800 nm). The transparency (T) of films was measured at 600 nm, and calculated as [18]:(5)$$T=\frac{A_{600}}{d}$$
where $A_{600}$ was the absorbance of film at 600 nm, *d* was the film thickness (mm).

#### 2.4.8. Mechanical Properties of Films

Tensile strength (TS) and percent elongation at break (E) of films were determined following the standard method D882. A Texture Analyzer Universal TA (Tenge instrument technology Co., Ltd., Shanghai, China) was used to determine TS and EAB of the films. The films with the size of 4 cm × 0.5 cm were stretched at the rate of 0.5 mms^−1^ until breaking with a 50 mm initial distance of separation. The analysis was done at 22 ± 1 °C and 50 ± 5% RH [2]. At least eight samples of each formulation were measured.

#### 2.4.9. Microstructure of Films

Film samples were mounted on the specimen holder using double-sided adhesive tapes. The method by Cao [19] was carried out with modification. The surface morphological was characterized by a scanning electron microscope (SEM) (Sigma 500, Carl Zeiss AG, Berlin, Germany) under high vacuum mode at an accelerating voltage of 10 kV.

### 2.5. Fresh-Keeping Properties of Films

Fresh blueberries were purchased from local market in Weihai, China. Fruits without visible mechanical damage or fungal infection were selected for study. Firstly, the fresh blueberries were randomly divided into three groups (ten blueberries per group). For each group, blueberries were placed in an aluminum trays (40 mm in diameter), the films were applied to the opening of the aluminum trays and tied with rubber bands. The samples were stored at room temperature for 5 days. The following parameters were monitored every day.

#### 2.5.1. Color of the Blueberry Samples

The surface color of blueberries were measured using a colorimeter. *a* and *b* parameters from CIE Lab scale were recorded and Hue angles (*h*°) index was calculated [20]:(6)$$h{}^{\circ}=\mathrm{arctg}\left(\frac{b}{a}\right)$$

#### 2.5.2. Weight Loss (WL) of the Blueberry Samples

The weight of the blueberries was monitored for 5 days. Weight loss was expressed as follows [21]:(7)$$WL=\frac{\left(m_{1}-m_{2}\right)}{m_{1}}\times 100\%$$
where, *m*_1_ was the initial weight of the blueberries, *m*_2_ was the weight of the blueberries after storage for a certain period.

#### 2.5.3. Total Soluble Solids (TSS) Content and pH of the Blueberry Samples

Blueberries were grinded and crushed to juice TSS was measured using a digital refractometer (TD-35; Top Instruments, Co., Ltd., Zhenjiang, China). The values were expressed as degree Brix (°B). pH values of the blueberry juice was determined with a pH meter (CRISON GLP21, Shinghai Shilu-Instruments, Shanghai, China) [20].

#### 2.5.4. Titratable Acidity (TA) Content of the Blueberry Samples

The samples for inspection was prepared as follows: the blueberry juice (as mentioned in Section 2.5.3) was diluted 10 times with distilled water and then the pH was adjusted to 8.1 with 0.10 mol/L NaOH [22]. The titrable acidity, expressed as g of citric acid per 100 g of fruit, was calculated as follow:(8)$$TA=\frac{V\times 0.1\times 0.064}{m}\times 100$$
where *V* was the volume of titration NaOH, 0.1 was the molarity of NaOH, 0.064 was the conversion factor for citric acid and *m* was the weight of blueberries.

#### 2.5.5. Total Anthocyanin Content of the Blueberry Samples

Blueberries (10 g) were mixed up and ground with 20 mL extraction solution (400 mL of ethanol, 100 mL of deionized water, and 5 mL of anhydrous acetic acid). After sanding in dark conditions for 60 min (at room temperature), a homogenized sample was obtained using centrifuge (TDL-80-2B, Huyueming Scientific Instruments Co., Ltd., Shanghai, China) at 3000 rpm for 15 min, the supernatants were collected to measure the anthocyanin content [22]. The result of total anthocyanin content was based on the absorbance of anthocyanins at 510 and 700 nm respectively in the buffer solution at pH 1.0 and pH 4.5, respectively. Data reported as milligrams of cyanidin-3-glucoside (C3G) equivalents per 100 g of fresh weight (FW) were calculated using the following equation:*A_tot_* = (*A*_510_−*A*_700_)*pH*1.0−(*A*_510_−*A*_700_)*pH*4.5(9)
where *(A*_510_ − *A*_700_*) pH 1.0* was the difference in absorbance at 510 and 700 nm of extraction in pH 1.0 buffer solution, *(A*_510_ − *A*_700_*) pH 4.5* was the difference in absorbance at 510 and 700 nm of extraction in pH 4.5 buffer solution. 

### 2.6. Statistical Analysis

All experiments were performed in triplicate. Data were expressed as mean ± standard deviation (SD) and evaluated statistically using statistical software SPSS Statistics 22.0. Figures were generated with Origin 9.0. The value of *p* < 0.05 was considered statistically significant.

## 3. Results and Discussion

### 3.1. Intermolecular Interaction between FG and OS

FT-IR spectra of FG, OS, FO3, and FO4 films were shown in Figure 1. Amide I and amide II bands were typical spectral features for gelatin, which located at approximately 1634 and 1548 cm^−1^, respectively. For the OS, a series of C-O and C-C vibrational peaks appeared at at 1185–878 cm^−1^, which indicated the formation and crystalline order of starch. The stretching vibration of alcohol hydroxyl groups for all spectra exhibited the peak at 3200–2900 cm^−1^ [23]. Comparison to OS and FODF spectra, the absorption of amide A at 3288 cm^−1^ slightly shifted toward lower wavenumbers (3284 cm^−1^) with the addition of OS, indicating the formation of hydrogen bonds between FG and OS [24]. This is in accordance with the work by Kuang Li [25], who found that the addition of chitosan in soy protein isolate films resulted in the shift of FT-IR absorption. Compared with FG and OS spectra, the peaks intensity of the FODF spectrum showed significantly decrease at 800 and 1750 cm^−1^, which corresponded to C-H and C=O vibration. This phenomenon confirmed that the reaction between amine groups and aldehydes might form the covalent cross-linking induced by Schiff’ base reaction [11,12].

### 3.2. Physico-Chemical Properties

#### 3.2.1. Color Values of Films

As an important index of films, the color was closely related to the appearance and consumer acceptance. [14]. The color values characterized with *L* and *ΔE* were presented in Table 1. As OS concentration increased, *L* value increased from 87.95 to 88.58. The incorporation of OS resulted in a slight decrease (less than 1) in *ΔE*, which inferred that the difference in color between film samples could hardly be distinguished by naked eyes.

#### 3.2.2. Optical Properties of Films

The light transmission of films was evaluated using UV-Vis spectrophotometer in the range of 200–800 nm. Table 2 demonstrated that all films exhibited excellent barrier properties in UV radiation range (200 and 280 nm). It might be associated with the presence of aromatic amino acids in gelatin, which were able to absorb radiation [26]. This result indicated that the gelatin films were able to retard lipid oxidation in food system induced by UV light. In the visible region (350–800 nm), the light transmission of all films showed an increasing tendency as the wavelength increased and all samples showed high light transmission. Moreover, the values did not have significant difference among the samples.

Transparency was a vital parameter in packaging field, especially for products containing photosensitive compounds [18]. The transparency values of all films were evaluated based on the absorbance at 600 nm (shown in Table 2). In general, the OS content did not significantly affect the T values of edible films. All samples showed the values ranged from 1.72–1.79.

As a result, the addition of OS hardly had any effect on the light barrier property and transparency of FODF.

#### 3.2.3. Moisture Contents of Films

The moisture content shows the hydrophilicity of films. [27]. As shown in Table 1, the MC of film samples ranged from 13.68% to 15.35%. As OS concentration increased, the MC of composite films showed a slight increase, followed by a decrease tendency. FO2 showed the highest MC (15.35%). The decrease in MC might be due to the formation of covalent linkages between FG and OS, which led to a decrease in the availability of hydroxyl and amino groups. As a result, the water absorption capacity of the polar groups in these matrices were limited [28]. Wei Li [29] also reported that the interactions between clove oil, chitosan, and the starch in composite films reducing the tendency of hydroxyl groups to interact with water, thus leading to a more hydrophobic matrix. Moreover, the excess OS might affect the FG molecular stretch to some extent, which would contribute to the decrease of water holding capacity of FG molecule [30].

#### 3.2.4. Barrier Properties of Films

The WVP, oil permeability, and OP values of different films were presented in Table 1. The lower WVP indicated the better performance on preventing water transfer [15]. As OS concentration increased, WVP of composite films showed a decreasing tendency, followed by a slight increase. The values ranged from 3.64 to 11.29 gm^−1^d^−1^MPa^−1^. FO3 showed the best water barrier properties, and FG film showed the worst. Oxygen permeability experiments were carried out on FG, FO3, and FO4 films which showed excellent water vapor barrier performance. FO3 showed the lowest oil permeability and OP values, while these of FG film were much higher.

This result could be attributed to the structure in FODF. FG-OS interpenetrating double network structure would be fabricated under thermal treatment, simultaneously, the cross-linked double network structure could be formed based on the Schiff’ base reaction. In general, the double network structure was responsible for the exceptional barrier properties of composite films, which could resist the diffusion of water, oil and oxygen. However, too much OS might break the stable structure and weaken the molecular force between the polymers [19], therefore, the parameters increased again.

#### 3.2.5. Mechanical Properties of Films

The mechanical properties of the edible films were characterized by TS and EAB (Figure 2). It can be deduced that the combination and interaction between matrix could result in the balanced tensile properties of films.

As OS concentration increased from 0% to 2.0% (w/v), TS enhanced by 57.41% (from 5.80 to 9.13 MPa). This could be attributed to the crosslinked network between FG and OS based on Schiff’ base reaction. As a result, the network density was improved effectively to resist the external stretching [31]. EAB is an important parameter for packing materials and larger EAB values indicated superior flexibility of films [2]. As shown, the presence of OS enhanced EAB and the highest value was obtained at 1.5% OS incorporated. The improvement in TS and EAB of the FODF might be related to the interaction between FG and OS, which were the result of the biological macromolecular material mediated by protein–protein and protein–polysaccharide interactions [32]. However, as OS concentration further increased, OS molecules might undermine free volume and molecular mobility, as a result, the flexibility of the chains would decrease.

#### 3.2.6. Appearance and Morphology of Films

FO3 and FO4 were adopted to observe the visual and microscopic structure because of their superior barrier and mechanical properties. Figure 3 shows the visual images and SEM graphs of FG, FO3, and FO4. It could be demonstrated that all films were transparent and visually compact. It could be noticed that the incorporation of OS brought out notable changes on the microstructure, the composite films with 1.5% (w/v) OS showed smooth and homogeneous microstructure containing few cracks or holes. While when a high concentration of OS (2.0%) was loaded, the surfaces of the films became rougher, which might affect their physical properties. Additionally, the surface microstructure of OS incorporated films showed the intactness surface, but the FG films demonstrated cavities. Hence, the incorporation of OS yielded superior film morphology to that of none addition. This phenomenon could be attributed to the formation of stable and uniform emulsion between FG and OS. As a result, it could be concluded that the presence of OS improved the morphology and structure of FG film noticeably.

### 3.3. Freeh-Keeping Properties of Films

The blueberries coated with FG, FO3 films, were kept at room temperature for 5 days and the uncoated ones was set as the control. Their changes in the color, WL, and physicochemical properties (TSS, pH, TA, and total anthocyanin content) were investigated as shown in Figure 4.

#### 3.3.1. Color Values of Blueberries

Figure 4A showed hue angles (*h*°) of blueberries during the 5 day storage period. At the fifth day, the *h*° were lower than the initial value, the FO3 coated samples showed the highest *h*° values. The decrease in *h*° of blueberries was mainly caused by the degradation of anthocyanins during ripening [20]. It indicated that FODF could retard the maturation of blueberries and slow down the decrease of *h*°.

#### 3.3.2. Weight Loss of Blueberries

As observed in Figure 4B, all samples showed the weight loss over storage period, which could be attributed to the water loss and dry matter consuming [33], while the water loss was mainly due to the stomata transpiration and direct evaporation through epidermal cell [34], and the dry matter consuming was caused by the metabolize process of fruits [35]. At the fifth day, the WL for the control samples (5.79%) was significantly higher than the FO3 coated one (3.93%). As seen, because of the physical barrier to moisture and oxygen diffusion, the edible films were effective in retrading the dehydration and metabolism during storage period [21].

#### 3.3.3. Total Soluble Solids Contents and pH

TSS contents were related to the fruit quality and consumer acceptability [36]. As shown in Figure 4C, TSS contents of all samples showed an increasing tendency in 5 days, particularly the uncoated samples showed the highest TSS (16.50° B). The increase in TSS was a result of the production of sucrose, glucose, and fructose during storage, which reflected the ripeness of the fruits [37].

As shown in Figure 4D, the pH values showed an increase tendency during storage period, at the fifth day, the lowest pH value (3.19) was obtained at FO3 coated samples. The increase in pH values was mainly due to the formation of alkaline autolysis compounds, which was commonly the symbol of ripeness [38]. Through these results, it can be concluded that the edible films were able to postpone the metabolism of blueberries and then prolong their storage period eventually.

#### 3.3.4. Titratable Acidity and Total Anthocyanin Content

The titratable acid was one of the important indexes to evaluate the quality of fruits [39]. The decrease was due to the organic acids consumption in the breathing process [40]. TA contents decreased during the storage period and the control blueberries showed the lowest TA values (0.65%) at the fifth day (Figure 4E). Thus, the edible films coating delayed TA losses in blueberries.

The changes in anthocyanin content was shown in Figure 4F. Blueberries are one of the richest sources of anthocyanins [22]. The anthocyanin content showed the decrease tendency for all blueberries. After 5 days of storage, the FO3 coated blueberries showed the highest anthocyanin content (38.20 mg _C3G_/100 g FW). The total anthocyanin content decreased with the ripening of blueberries, a low pH is also the external factor that favor the stability of anthocyanin pigments. As seen in Figure 4D, the FO3 coated blueberries remained the lowest pH values during storage period which was favorable to the stability of anthocyanin content.

As mentioned above, the edible films packing, especially the FO3, could postpone the ripening of blueberries, thus, it could favor the stability of hue angels, weight and physicochemical properties (including TSS, pH, TA, and total anthocyanin content). The similar results could be seen in the research of Cheng Zhang [41], who used nano-Ag-polylactic acid composite film to package strawberries and measured the physicochemical properties to assess freshness. They found that the packaging film could effectively preserve freshness of strawberries. The presence of oxygen could promote the ripening process, the FO3 had better oxygen barrier performance, hence, it could effectively prevent partial oxygen passing through the films. This was consistent with the results of barrier properties as shown in Table 1.

## 4. Conclusions

In this study, OS was incorporated at different ratios into the film forming solution based on FG. The composite films were developed by thermal treatment and Schiff’ base reaction. The formed films possessed a homogeneous network with good distribution of OS throughout the matrix. OS loading was found to enhance the water binding capacity, mechanical behavior (tensile properties and flexibility), and increase the light transmission properties of films. Furthermore, FT-IR spectra indicated the covalent linkage and the great compatibility between FG and OS. According to SEM microscopy, FODF showed smooth surface indicating that OS could improve the microscopic appearance of FG. The optimized conditions to develop FODF was to be found at the FG/OS ratio of 5.0/1.5, which showed the lowest WVP (3.64 gcm^−1^d^−1^MPa^−1^), oil permeability (0.88 gm^−2^h^−1^), OP (10.98 cm^3^/m^2^ d·0.1Mpa). In fresh-keeping experiments, FO3 was effective in retrading the dehydration and postponing the metabolism of blueberries, which indicated the best fresh-keeping performance. Therefore, it could conclude that FODF have great potential to be further explored as novel material for edible packaging applications.

## Figures and Tables

**Figure 1 polymers-11-02065-f001:**
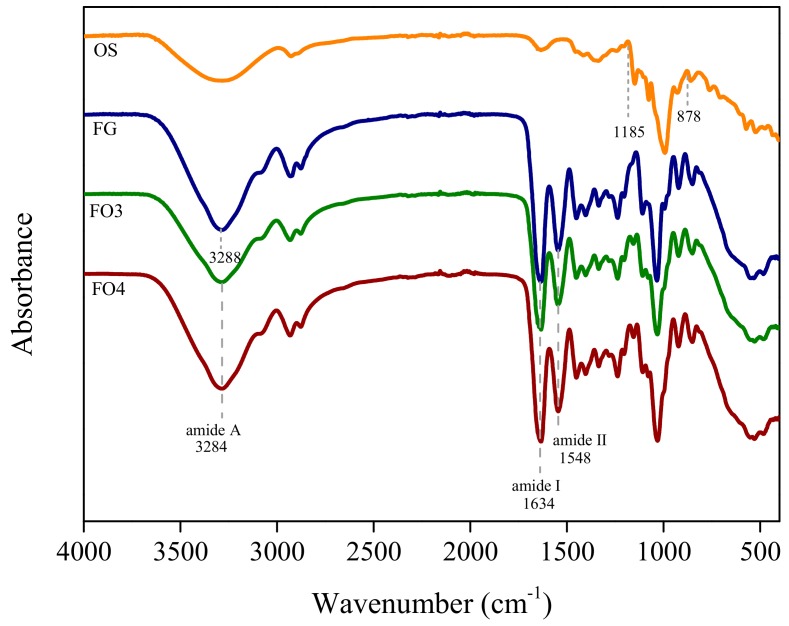
FT-IR spectra of different samples.

**Figure 2 polymers-11-02065-f002:**
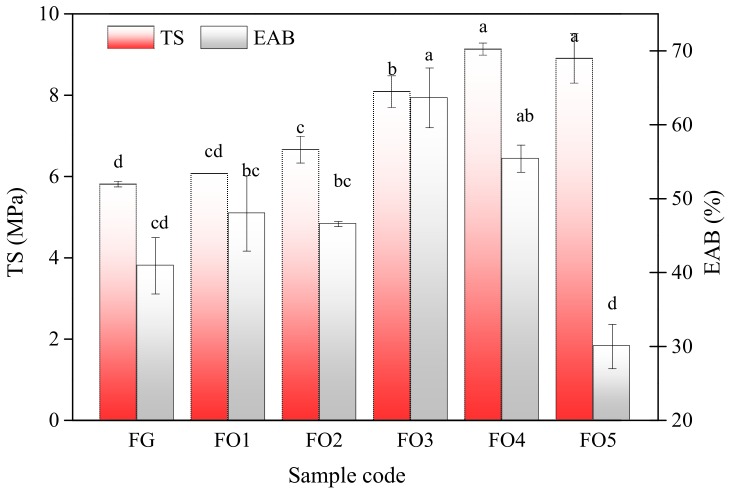
Tensile strength and elongation at break of films. Different letters (a–c) in the same column indicate significant differences (*p* < 0.05).

**Figure 3 polymers-11-02065-f003:**
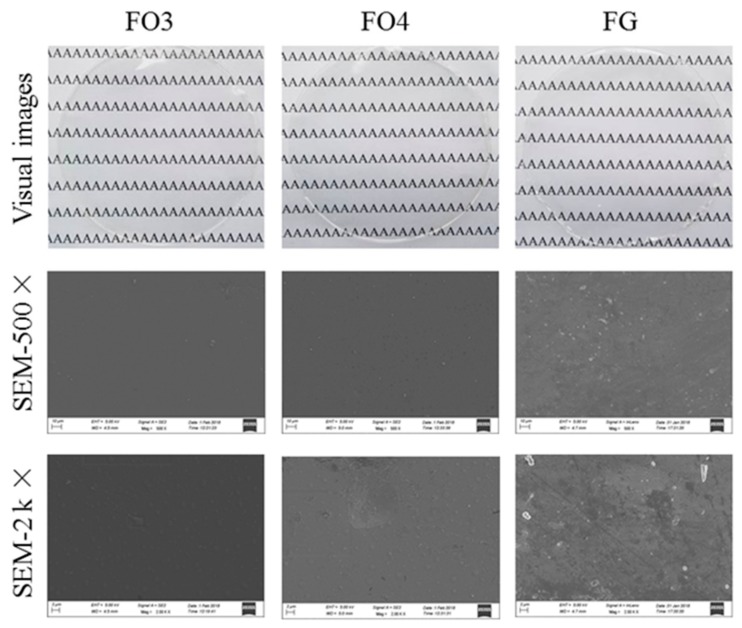
Visual and microscopic graphs of films.

**Figure 4 polymers-11-02065-f004:**
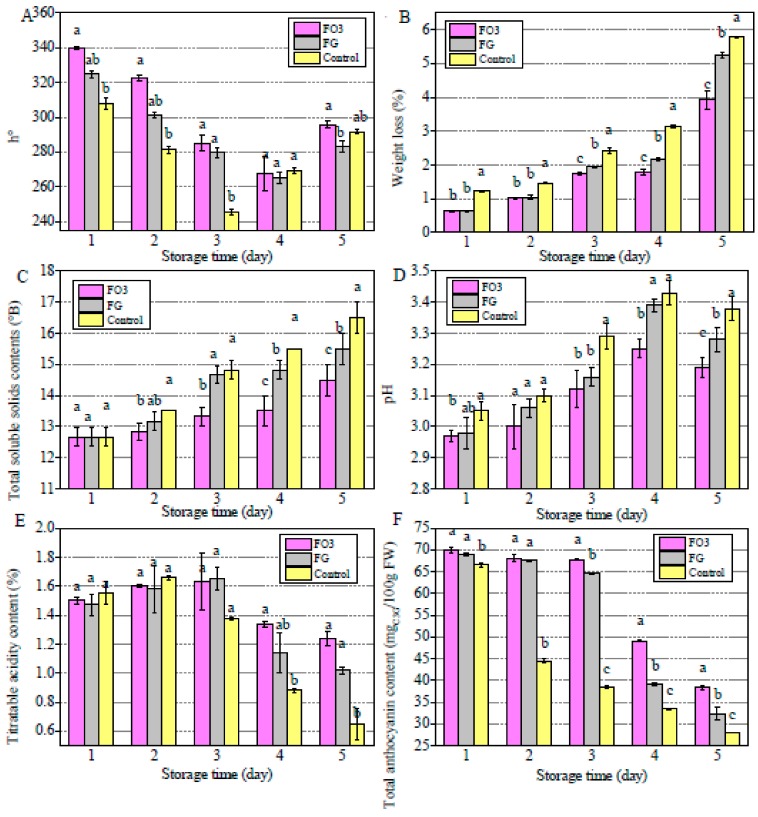
Effects of different preservative film on properties of blueberries (**A**) *h*° values, (**B**) weight loss, (**C**) total soluble solids contents, (**D**) pH, (**E**) titratable acidity content, and (**F**) total anthocyanin content. Different letters (a–c) in the same column indicate significant differences (*p* < 0.05).

**Table 1 polymers-11-02065-t001:** The effect of oxidized starch (OS) concentration on the color values, moisture content (MC), and barrier properties (water vapor permeability (WVP), oil permeability, and oxygen permeability (OP)) of films.

Sample Code	*L*	*ΔE*	MC/%	WVP/gm^−1^d^−1^MPa^−1^	Oil Permeability/gm^−2^h^−1^	OP/cm^3^m^−2^ d^−1^0.1MPa^−1^
FG	88.79 ± 0.26 ^a^	0	14.47 ± 0.03 ^c^	11.29 ± 0.99 ^a^	3.73 ± 0.45 ^a^	18.21 ± 0.01 ^a^
FO1	87.95 ± 0.78 ^b^	0.91 ± 0.58 ^a^	14.82 ± 0.01 ^b^	9.91 ± 0.78 ^b^	3.54 ± 0.29 ^a^	/
FO2	87.96 ± 0.18 ^b^	0.78 ± 0.17 ^a,b^	15.35 ± 0.09 ^a^	9.22 ± 0.31 ^b^	2.65 ± 0.29 ^b^	/
FO3	88.30 ± 0.10 ^a,b^	0.49 ± 0.06 ^a,b^	14.54 ± 0.03 ^c^	3.64 ± 0.03 ^d^	0.88 ± 0.01 ^d^	10.98 ± 0.68 ^b^
FO4	88.45 ± 0.26 ^a,b^	0.57 ± 0.15 ^a,b^	14.48 ± 0.12 ^c^	4.59 ± 0.42 ^d^	1.67 ± 0.17 ^c^	11.23 ± 1.04 ^b^
FO5	88.58 ± 0.03 ^a,b^	0.41 ± 0.06 ^b^	13.68 ± 0.04 ^d^	8.03 ± 0.44 ^c^	2.36 ± 0.01 ^b^	/

Values are mean ± standard deviation. Different letters (a–d) in the same column indicate significant differences (*p* < 0.05).

**Table 2 polymers-11-02065-t002:** The effect of OS concentration on the light transmission and transparency of film.

		FG	FO1	FO2	FO3	FO4	FO5
Light transmission (%) at different wave number (nm)	200	0.12 ± 0.01 ^a^	0.13 ± 0.01 ^a^	0.13 ± 0.01 ^a^	0.12 ± 0.02 ^a^	0.11 ± 0.02 ^a^	0.10 ± 0.02 ^a^
	280	44.20 ± 7.58 ^a^	47.57 ± 3.70 ^a^	47.80 ± 7.31 ^a^	46.10 ± 8.92 ^a^	43.97 ± 2.55 ^a^	32.47 ± 6.89 ^a^
	350	91.93 ± 1.10 ^a^	93.37 ± 0.81 ^a^	91.93 ± 0.72 ^a^	92.23 ± 0.40 ^a^	92.10 ± 0.78 ^a^	88.80 ± 1.65 ^b^
	400	93.20 ± 0.87 ^b^	94.30 ± 0.17 ^a^	92.93 ± 0.40 ^b^	93.30 ± 0.20 ^b^	93.43 ± 0.38 ^a,b^	91.37 ± 0.67 ^c^
	500	92.90 ± 0.66 ^b^	93.77 ± 0.35 ^a^	92.70 ± 0.17 ^b,c^	93.27 ± 0.25 ^a,b^	93.40 ± 0.35 ^a,b^	92.17 ± 0.76 ^c^
	600	93.57 ± 0.25 ^a^	92.70 ± 0.53 ^a^	92.93 ± 0.70 ^a^	92.93 ± 0.93 ^a^	93.33 ± 0.15 ^a^	93.57 ± 0.45 ^a^
	700	93.20 ± 0.95 ^a^	93.67 ± 0.15 ^a^	93.00 ± 0.60 ^a^	93.57 ± 0.06 ^a^	93.67 ± 0.35 ^a^	92.77 ± 0.55 ^a^
	800	93.23 ± 0.87 ^a^	93.77 ± 0.12 ^a^	93.07 ± 0.38 ^a^	93.30 ± 0.10 ^a^	93.57 ± 0.45 ^a^	92.83 ± 0.57 ^a^
T		1.79 ± 0.02 ^a^	1.72 ± 0.04 ^a^	1.73 ± 0.05 ^a^	1.73 ± 0.07 ^a^	1.77 ± 0.45 ^a^	1.78 ± 0.04 ^a^

Values are mean ± standard deviation. Different letters (a–c) in the same column indicate significant differences (*p* < 0.05).

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
