# Peer review of "Development and Properties of Fish Gelatin/Oxidized Starch Double Network Film Catalyzed by Thermal Treatment and Schiff’ Base Reaction"

_polymers, 2019, doi:10.3390/polym11122065_

Round 1

Reviewer 1 Report

Comments to the Author

The work reported in this manuscript is interesting and well presented. However, it needs improvements before the acceptance. The work requires major revision. Some comments are:‎

Comment 1: The duplication of the article reached *30%*, while the standard of the journal is lower than *25%*. Please refer to the duplication report attached and reduce the duplication rate.

Comment 2: The SEM images are not clear and very poor visibility, therefore it needs to be re measure the SEM analysis of all films with high resolution. Also, need to add the more explanation of all films surface morphology.

Comment 3: In FT-IR spectra (Fig. 1), the authors provided the spectra of FG, OS, FO3 and FO4. The remaining films FTIR spectra not provided in this figure, so the authors need to provide the all films FTIR spectra in order to check the covalent linkage and the great compatibility between FG and OS induced by Schiff’ base reaction.

Comment 4: In references, the authors mentioned some places with full journal name and some places with abbreviation of the journal. Therefore, they need to revise the references according to the journals standard format.

Reviewer 2 Report

The paper reads very well and authors have done extensive characterization tests to validate the suitability of using their films in food packaging applications. I think the paper can be published but I have a few concerns and suggestions:

1- There is a paper with almost a same title in a conference book. The 20th Gums & Stabilisers for the Food Industry Conference. What's the difference between these two works and I think the title of the paper has to be changed due to this reason. 

2- Authors are keep reporting some numbers and values but we don't know how they benchmarked their properties with the values for a real food package which is already used in the market. How can readers know that those numbers are matched with the standards and how should we know that these properties are suitable for a food package. I suggest authors compare their results with other papers as well as real food package films. 

3- I think authors should also cite related papers from other journals specially "polymers". There are a lot of interesting papers on food packaging published in "Polymers". At this moment most of the resources are focused on one specific journal. 

Round 2

Reviewer 1 Report

I believe the manuscript has been significantly improved and now warrants publication in Polymers